# Gene Editing of the Catfish Gonadotropin-Releasing Hormone Gene and Hormone Therapy to Control the Reproduction in Channel Catfish, *Ictalurus punctatus*

**DOI:** 10.3390/biology11050649

**Published:** 2022-04-24

**Authors:** Guyu Qin, Zhenkui Qin, Cuiyu Lu, Zhi Ye, Ahmed Elaswad, Max Bangs, Hanbo Li, Yiliu Zhang, Yingqi Huang, Huitong Shi, Kamal Gosh, Nermeen Y. Abass, Khoi Vo, Ramjie Odin, William S. Bugg, Nathan J. C. Backenstose, David Drescher, Zachary Taylor, Timothy Braden, Baofeng Su, Rex A. Dunham

**Affiliations:** 1School of Fisheries, Aquaculture and Aquatic Sciences, Auburn University, Auburn, AL 36849, USA; gzq0002@auburn.edu (G.Q.); qinzk@ouc.edu.cn (Z.Q.); czl0094@auburn.edu (C.L.); zzy0008@auburn.edu (Z.Y.); ahe0001@auburn.edu (A.E.); maxbangs@gmail.com (M.B.); hzl0026@auburn.edu (H.L.); yzz0136@auburn.edu (Y.Z.); yzh0065@auburn.edu (Y.H.); hzs0052@auburn.edu (H.S.); kgosh@langston.edu (K.G.); n.y.abass@alexu.edu.eg (N.Y.A.); kmv0005@auburn.edu (K.V.); ryo0001@auburn.edu (R.O.); wsb0015@auburn.edu (W.S.B.); njb0012@auburn.edu (N.J.C.B.); ddrescher132@gmail.com (D.D.); zat0005@auburn.edu (Z.T.); dunhara@auburn.edu (R.A.D.); 2Ministry of Education Key Laboratory of Marine Genetics and Breeding, College of Marine Life Sciences, Ocean University of China, Qingdao 266003, China; 3Department of Animal Wealth Development, Faculty of Veterinary Medicine, Suez Canal University, Ismailia 41522, Egypt; 4Department of Biological Science, Florida State University, Tallahassee, FL 32304, USA; 5College of Fisheries and Life Science, Shanghai Ocean University, Shanghai 201306, China; 6Department of Agriculture and Natural Resources, Langston University, Langston, OK 73050, USA; 7Department of Agricultural Botany, Faculty of Agriculture Saba-Basha, Alexandria University, Alexandria 21531, Egypt; 8College of Fisheries, Mindanao State University-Maguindanao, Maguindanao 9601, Philippines; 9Department of Biological Sciences, University of Manitoba, Winnipeg, MB R3T 2N2, Canada; 10Department of Biological Sciences, State University of New York at Buffalo, Buffalo, NY 14228, USA; 11Muckleshoot Indian Tribe, Keta Creek Hatchery, 34900 212th Ave SE, Auburn, WA 98092, USA; 12Department of Anatomy, Physiology and Pharmacology, Auburn University, Auburn, AL 36849, USA; bradetd@auburn.edu

**Keywords:** channel catfish, transcription activator-like effector nucleases, catfish gonadotropin-releasing hormone, hormone therapy

## Abstract

**Simple Summary:**

Gonadotropin-releasing hormone (GnRH) plays a pivotal role in fish reproduction. In the present study, transcription activator-like effector nuclease (TALEN) plasmids targeting the catfish gonadotropin-releasing hormone (cfGnRH) gene were delivered into fertilized eggs with double electroporation to sterilize channel catfish (*Ictalurus punctatus*). After low fertility was observed, application of luteinizing hormone-releasing hormone analog (LHRHa) hormone therapy resulted in good spawning and hatch rates for mutants (individuals with human-induced sequence changes at the cfGnRH locus). Gene editing of channel catfish for the reproductive confinement of gene-engineered, domestic, and invasive fish to prevent gene flow into the natural environment appears promising.

**Abstract:**

Transcription activator-like effector nuclease (TALEN) plasmids targeting the channel catfish gonadotropin-releasing hormone (cfGnRH) gene were delivered into fertilized eggs with double electroporation to sterilize channel catfish (*Ictalurus punctatus*). Targeted cfGnRH fish were sequenced and base deletion, substitution, and insertion were detected. The gene mutagenesis was achieved in 52.9% of P_1_ fish. P_1_ mutants (individuals with human-induced sequence changes at the cfGnRH locus) had lower spawning rates (20.0–50.0%) when there was no hormone therapy compared to the control pairs (66.7%) as well as having lower average egg hatch rates (2.0% versus 32.3–74.3%) except for one cfGnRH mutated female that had a 66.0% hatch rate. After low fertility was observed in 2016, application of luteinizing hormone-releasing hormone analog (LHRHa) hormone therapy resulted in good spawning and hatch rates for mutants in 2017, which were not significantly different from the controls (*p* > 0.05). No exogenous DNA fragments were detected in the genome of mutant P_1_ fish, indicating no integration of the plasmids. No obvious effects on other economically important traits were observed after the knockout of the reproductive gene in the P_1_ fish. Growth rates, survival, and appearance between mutant and control individuals were not different. While complete knock-out of reproductive output was not achieved, as these were mosaic P_1_ brood stock, gene editing of channel catfish for the reproductive confinement of gene-engineered, domestic, and invasive fish to prevent gene flow into the natural environment appears promising.

## 1. Introduction

Channel catfish (*Ictalurus punctatus*) and their hybrids comprise greater than 60% of aquaculture production within the US [1,2]. The catfish industry has been severely challenged in recent years as a result of higher operating costs, diseases, and competition from cheaper imported frozen catfish (*Ictalurus* and *Pangasianodon hypophthalmus*) [3].

A potential future approach to help address this problem is the utilization of genetically engineered catfish to increase yield, enhance disease resistance, improve survival ability in extreme environments and produce specific proteins [4,5,6,7]. However, concerns have been expressed regarding the potential ecological and genetic effects of these fish [8]. An effective fish sterilization technology could prevent the environmental risk of transgenic fish.

Common methods used to genetically sterilize fish include triploid induction, genetically monosex populations, and interspecific hybridization. However, these techniques do not always result in 100% sterility [6], and they require fertile diploid broodstock; thus, environmental risk cannot be eliminated. Additionally, commercial-scale triploidy in ictalurid catfish and other species would be labor intensive, and triploidy usually decreases the overall performance of fish [9]. Transgenic sterilization technology has been developed to knock down genes essential for reproduction [10,11]. However, transgenesis could result in possible adverse pleiotropic effects, such as reduced growth rate [12]. Targeted genome editing technologies, including zinc finger nucleases (ZFNs), transcription activator-like effector nucleases (TALENs) and clustered regularly interspaced short palindromic repeats (CRISPR), can be utilized to mutate reproductive genes to sterilize catfish and prevent the potential risks of genetically modified fish. ZFN technology has been used to knock out the luteinizing hormone (LH) gene to effectively create gene-edited sterile channel catfish [13,14]. TALENs have been used to knock out the LH gene and follicle-stimulating hormone (FSH) genes to inhibit gonadal development and ovulation in zebrafish (*Danio rerio*) [15].

Gonadal maturation in teleost fishes is principally controlled by the hypothalamus–pituitary–gonad (HPG) axis. Gonadotropin-releasing hormone (GnRH) plays a pivotal role in adjusting the differentiation of the gonad via the HPG axis [16]. GnRH is a releasing hormone secreted from the hypothalamus by neurosecretory cells. In teleost fishes, GnRH-expressing neurons are distributed among two or three distinct GnRH populations within the brain [17]. GnRH can stimulate the synthesis and release of pituitary gonadotropin, followed by stimulating the secretion of steroid hormones. Additionally, GnRH might be involved in regulating the reproductive behavior of fish [18]. GnRH can activate the secretion of other pituitary hormones, which include growth hormone [19], prolactin, and somatolactin [20]. Siluriformes possess two forms of GnRH—chicken GnRH-ΙΙ (cGnRH-ΙΙ) and catfish GnRH (cfGnRH). The former is commonly accepted as a neuromodulator or plays a role in sexual behavior, while the latter is the hypothalamic GnRH form and considered to be responsible for the release of gonadotropin, playing a critical role in final gamete maturation in catfish (*Clarias gariepinus*) [21,22]. The knockout of the cfGnRH gene should restrain the synthesis and release of pituitary gonadotropin, followed by the repression of the secretion of steroid hormones, including estradiol and testosterone. The knockout of the cfGnRH gene may also change the reproductive behavior of fish by increasing GnRH immunoreactive cell numbers and fibers [18].

All vertebrate groups share a common structure of GnRH with three introns and four exons. The typical structure of the GnRH precursor protein consists of an untranslated region and a signal peptide of about 23 amino acids, a decapeptide followed by a 3-amino acid cleavage site, and a GnRH-associated peptide (GAP) followed by C-terminal region of total about 60 amino acids [23]. Nikolics et al. [24] reported that a complete GAP peptide has dual activities, one inhibits the activity of prolactin and the other stimulates the releasing of gonadotropin-releasing in cultured rat pituitary cells. Millar et al. [25] stated that GAP could stimulate the release of gonadotropins (LH and FSH) in vitro. Andersen and Klunglan [26] reviewed the function of GAP and GnRH and concluded that GAP might play a role in conferring the structure and stability of prepro-GnRH. Wetsel and Srinivasan [27] proposed that GAP serves as a linker to connect the decapeptide by the amide donor glycine and the dibasic residues lysine and arginine so that pro-GnRH can be guided and routed out of the endoplasmic reticulum into the secretory pathway. A complete amino acid constitution of GAP peptide is required to exert full activity in conjugate with GnRH decapeptide.

GnRH is responsible for the release of gonadotropins, FSH and LH. FSH and LH vitalize germ cell development via activation of the FSH receptor and LH receptor, partly by stimulating the production of sex steroids in gonadal somatic cells and by releasing gonadal paracrine growth factors that manage the development of germ cells. These gonadal sex steroids and growth factors consecutively provide positive or negative feedback to the brain and pituitary to modulate FSH and LH production and secretion to complete the HPG axis [16]. FSH plays a major role in the earlier stage of the reproductive cycle by activating steroidogenesis, vitellogenesis, and spermatogenesis, while LH plays a vital role in the later stage on stimulating spermiation as well as ovulation and the maturation of oocytes [28].

The objectives of this study were to use TALENs to induce mutations that would block the function of the cfGnRH gene in order to sterilize channel catfish, and then restore the fertility of gene-edited sterile channel catfish through hormone therapy. Mutants in this study are individuals with human-induced sequence changes at the cfGnRH locus. Specific objectives included the knockout of the cfGnRH gene via double electroporation of TALEN plasmids into sperm and early embryos, evaluation of the reproductive performance of mutant catfish, restoration of reproduction in mutant fish via LHRHa treatment, verification of the germline transmission of mutations to future generations, and assessment of pleiotropic effects of gene knockout on survival, hatch rate and growth rate in P_1_ and F_1_ fish.

## 2. Materials and Methods

### 2.1. Construction of Plasmids

TALENs are proteins selected from the bacteria *Xanthomonas*. *Xanthomonas* transcription activator-like effector nuclease (XTNTM) plasmids were designed and assembled by Transposagen Company (Lexington, KY, USA). Each XTN was cloned into the SQT281 vector in the TAL repeat region and driven by the cytomegalovirus (CMV) promoter, T7 promoter, then followed by FokI nuclease domain (Appendix A). The TALEN target sites were designed to be located at exon3, which is a GAP region of the cfGnRH gene (GenBank Accession No. XM_017468372.1). Each TALEN comprised an 18 bp (5′-TTCACCTCGGAATAAACT-3′) left DNA-binding site, or an 18 bp (5′-TGAGCTGTGCACCAGCAG-3′) right DNA-binding site.

### 2.2. TALEN Plasmid Replication, Extraction and Dilution

The left and right TALEN plasmids were individually transformed into One Shot^®^ TOP10 Competent Cells (Invitrogen, Grand Island, NY, USA), following the transformation procedures from the manufacturer. A total of 100 µL of each transformation mix vial was spread on the LB agar plate with 100 ppm ampicillin. A single colony was picked from each plate and then cultured in 400 mL LB broth with 100 ppm ampicillin. Then, the TALEN plasmids were extracted with the IsoPure Plasmid Maxi ΙΙ Prep Kit (Denville, Holliston, MA, USA). DNA agarose gel electrophoresis and spectrophotometry were then used to check the quantity and quality of the TALEN plasmids. Equal amounts of both left and right TALEN plasmids were mixed together and separated into 2 sets. Each TALEN plasmid set was prepared for the purpose of double electroporation. One set of TALEN plasmids were diluted with 2 mL saline (0.9% NaCl) to the final concentration of 25 μg/mL for the first electroporation of sperm. The other set of TALEN plasmids were diluted with 5 mL TE buffer (5 mM Tris-HCl, 0.5 mM EDTA, pH = 8.0) for the second electroporation.

### 2.3. Experimental Brood Stock and Gametes

Sexually mature females of Kansas Random strain of channel catfish were selected and implanted with luteinizing hormone-releasing hormone analogue (LHRHa) Reproboost^®^ Implants (Center of Marine Biotechnology, Baltimore, MD, USA) with a dose of 90 µg/kg. All fish showed outstanding secondary sexual characteristics. Females were placed in spawning bags at least 15 cm under water in a flow-through tank with continuous 27 °C water flow and aeration. Ovulation was checked 36 h after implantation. Females were anesthetized in 100 ppm tricaine methane sulfonate (MS-222) in which the pH was adjusted to 7.0 using sodium bicarbonate, then were rinsed in pond water, and dried with a clean towel. Eggs of two anesthetized ovulating females were hand stripped into dried stainless-steel pie pans greased with vegetable shortening. Two-hundred eggs were collected for each experimental group and control group. Two Kansas Random strain males were euthanized by a percussive blow to the head followed by pithing. Mixed testes were placed into a mesh strainer and macerated manually into saline (0.9% NaCl) to collect sperm [29].

### 2.4. Fertilization, Double Electroporation and Embryo Incubation

The double electroporation was performed according to Dunham et al. [30]. Briefly, the first electroporation was conducted on the collected sperm homogenate with the TALEN saline solution using a Baekon 2000 macromolecule transfer system (Baekon, Inc., Saratoga, CA, USA) with parameters of 6 kV, 2^7^ pulses, 0.8 s burst, 4 cycles and 160 µs [31]. Then, two hundred prepared eggs were artificially fertilized with the electroporated sperm. After 1 h, the embryos were gathered and immersed in the TALEN TE buffer solution for 10 min. Then, the second electroporation was conducted with the same procedure. The control group was generated following the same protocols, but in the absence of TALEN plasmids. Embryos were then moved into 10 L tubs with Holtfreter’s solution containing 10 ppm doxycycline and incubated statically until hatch. Dead embryos were removed daily before Holtfreter’s solution was changed.

### 2.5. Sample Collection and Mutation Detection for P_1_ Generation

Embryos were cultured for 6–7 days until hatch in tubs at 27 °C and then transferred into a recirculating system. Fish in indoor circulating systems were cultured at oxygen levels above 5.0 ppm and un-ionized ammonia below 2 ppm, nitrite 0 ppm, pH between 6.8 and 7.8, hardness above 30 ppm and chlorine 0 ppm. Brine Shrimp (Artemia) Eggs (Carolina Biological, Burlington, NC, USA) were hatched in salt water and immediately fed to the fry 4 times a day. After one month, fry were fed Purina^®^ AquaMax^®^ Fry Powder (Purina, St. Louis, MO, USA) three times a day. As fry grew, they were then fed Purina^®^ AquaMax^®^ Fry Starter 100. After further growth, fry were moved to 90 L tank at a stocking density of 500 fish per tank and fed Purina^®^ AquaMax^®^ Fry Starter 200 and 300 three times a day. Any excess feed was removed from the tanks and feeding rates were reduced. Feeding rates were also reduced as the water temperature lowered. A 50% protein powdered fry starter was fed to fry, 36–48% protein pellets for small fingerlings and 32–36% protein floating fingerling feed for fingerlings. After 6 months, fishes reached around 100 g in body weight, and the genomic DNA was extracted from the pelvic fin and barbel samples. Samples were digested with cell lysis buffer and proteinase K following the protocols by Cheng et al. [32]. DNA agarose gel electrophoresis and spectrophotometry were then used to check the quantity and quality of the genomic DNA. Then, Roche Expand High Fidelity Plus PCR System (Roche, Indianapolis, IN, USA) was used to amplify the channel catfish cfGnRH gene on these DNA samples. The following primers were utilized: forward sequence 5′-ATGGATGCTGTCTTTGTTTTCC-3′; reverse sequence 5′-CCACACGAAATAAAGGCAA AG-3′ (Table 1), and the amplification procedure was: initial denaturation for 2 min at 94 °C, followed by 29 cycles of 94 °C for 30 s, 60 °C for 30 s and 72 °C for 40 s, and the final elongation for 10 min at 72 °C.

Gene mutations were detected using Surveyor^®^ Mutation Detection Kit for Standard Gel Electrophoresis prior to sequencing (Integrated DNA Technologies, Coralville, IA, USA). The processes were as follows: first, the PCR products were hybridized using a PCR machine to form a heteroduplex at 95 °C for 10 min, 85 °C for 1 min (Ramp 2 °C/s), 75 °C for 1 min (Ramp 0.3 °C/s), 65 °C for 1 min (Ramp 0.3 °C/s), 55 °C for 1 min (Ramp 0.3 °C/s), 45 °C for 1 min (Ramp 0.3 °C/s, 35 °C for 1 min (Ramp 0.3 °C/s) and 25 °C for 1 min (Ramp 0.3 °C/s). Then, a nuclease reaction system, which included DNA hybridization system, 150 mM MgCl_2_, Surveyor enhancers S, and Surveyor nucleases, was established and incubated in a PCR machine at 42 °C for 1 h. Finally, the 2% UltraPure Agarose-1000 (Invitrogen, Grand Island, NY, USA) was used to perform electrophoresis to test the mutation.

### 2.6. TA Clone and Sequencing

To confirm the presence of the mutations, several DNA samples that showed multiple DNA fragments after Surveyor nuclease cleavage were randomly selected and sequenced. Roche high-fidelity PCR amplification targeting cfGnRH gene was performed with the sample DNA. Then, TOPO^®^ TA Cloning^®^ kit (Thermo Fisher Scientific, Waltham, MA, USA) was used to set up a 6 µL TOPO cloning system, which included 4 µL amplified DNA product, 1 µL salt solution, and 1 µL TOPO vector. The system was mixed and incubated at room temperature (22–23 °C) for 5 min, followed by competent cell transfection. First, 2 µL of the cloning system was added into one vial of One Shot^®^TOP10 competent cells (Thermo Fisher Scientific, Waltham, MA, USA) and put on the ice for 30 min. Then, 42 °C heat shock was implemented using a PCR machine for 30 s. After that, the tube was placed on ice immediately. Then, it was mixed with 250 µL LB broth in a 1.5 mL tube, placed in the 37 °C shaker and agitated at 200 rpm for 1 h. Next, 50 and 100 µL mixtures from the same transformation were spread on the LB agar plates and placed in the 37 °C incubator overnight. Ten monoclonal colonies were picked from each plate, mixed with 400 mL LB broth and placed in the 37 °C incubator overnight. The amplified bacterial solution was dispensed into a 96-well plate and sent to the Eurofins Genomics Company (Louisville, KY, USA) for sequencing. Finally, sequencing results were analyzed using the T-Coffee tool.

### 2.7. Plasmid Integration Inspection

PCR was conducted to determine if plasmids were integrated into the genome or persisted in the cytoplasm. Two pairs of primers targeting cfGnRH gene were designed to detect the plasmid DNA in mutant fish. The amplification regions were the CMV promoter region and the TAL repeats region, respectively. They were amplified using the primers shown in Table 1. The amplification procedure was as follows: initial denaturation for 2 min at 94 °C, followed by 29 cycles of 94 °C for 30 s, 60 °C for 30 s and 72 °C for 40 s, and a final elongation for 10 min at 72 °C. The results were generated using electrophoresis.

### 2.8. Reproductive Evaluation

The fingerlings were transferred to 0.04-hectare pond or a recirculating system when they reached 15–20 cm, 300–500 g, for further growth and maturation. Fish in ponds were cultured under oxygen levels above 3.0 ppm, un-ionized ammonia below 2 ppm, nitrite 0 ppm, pH between 6.8 and 7.8, and hardness above 30 ppm. A 28–32% protein catfish feed was fed to food fish, and 32–36% protein brood stock feed for brood stock. Usually, fish were fed ad-libitum. Mutant fish were selected based on Surveyor results. Mating experiments began on 5 June 2016. The further sequencing investigation was conducted to analyze the phenotype observed in 2016. There were eight mutant (M) males and five mutant females selected to mate with wild-type (Wt) females and males (Wt♀ × M♂, M♀ × Wt♂). There were five pairs of control of Kansas Random strain of channel catfish selected for natural spawning without LHRHa implantation. These fish were paired in 48 cm × 36 cm × 21 cm aquaria with constant water flow and compressed air for aeration. All fish showed outstanding secondary sexual characteristics. The sexually mature males had well-developed head muscles, large and reddish genital papillae and dark mottling in the lower jaw and abdomen. The sexually mature females exhibited fuller, rounder abdomen and reddening genital area compared to immature females.

For testing the fertility of mutant males, only the wild-type females paired with the mutant males in aquaria were implanted with 75 μg/kg LHRHa for induced spawning. Then, the wild-type females that did not spawn within five days were implanted again with 70 μg/kg LHRHa. If the females did not spawn after two implants, they were replaced by other wild-type females twice and the hormone implantation for laying eggs. For testing the fertility of mutant females, the pairs were given 12 days to spawn naturally with the wild-type males and without LHRHa implantation.

### 2.9. Hormone Therapy

Sexually mature Kansas Random channel catfish (mutant) were implanted with LHRHa to restore the fertility. In 2016, three mutant males and three mutant females were paired, and both were implanted with 90 μg/kg LHRHa (M♀ × M♂). After nine days, the three pairs that had not spawned were implanted with 90 μg/kg LHRHa again.

After establishing infertility in 2016, hormone therapy was used in 2017. There were a total of five mutant males and six mutant females with outstanding secondary sexual characteristics selected to mate (M♀ × M♂). One male was mated with 2 consecutive females. Firstly, five females and five males were paired and injected with 100 μg/kg LHRHa underneath the pelvic fins. The LHRHa used for injection was made of LHRHa powder and dissolved in saline. It was the same formula of LHRHa just in liquid form instead of the implant form. The first spawned female was replaced by another sexually mature female fish, which was injected with 100 μg/kg LHRHa solution. If females and males did not spawn within five days, they were injected again with 80 μg/kg LHRHa. Three control pairs of channel catfish Kansas Random strain were selected for natural spawning without hormone injection. And other seven pairs of control were selected for spawning with 90 μg/kg LHRHa solution.

### 2.10. F_1_ Fish Culture

Egg masses were placed in baskets in hatching troughs with constant water flow and aeration (above 6 ppm). Calcium chloride solution was continually dripped into the trough to ensure ≥40 ppm calcium hardness at 27 °C. Eggs were gently agitated with a paddlewheel beginning 2 h after spawn collection. The egg masses were prophylactically treated with 100 ppm formalin or 32 ppm copper sulfate every 8 h to avoid fungus [33]. The treatments were terminated 12 h before hatch.

Catfish embryos began hatching in seven days with a water temperature between 26 and 28 °C. They consumed their yolk sac and began the swim-up stage three days post hatching. They were then fed Brine Shrimp (Artemia) Eggs (Carolina Biological, Burlington, NC, USA) three times a day and stocked into a recirculating system with densities of 1000 fish per 90 L tank and 100 fish per 60 L aquaria of each family. After one month, fry were fed Purina^®^ AquaMax^®^ Fry Powder (Purina, St. Louis, MO, USA) three times a day. As fry grew, they were then fed Purina^®^ AquaMax^®^ Fry Starter 100. After further growth, fry were moved to 90 L tank at a stocking density of 500 fish per tank and fed Purina^®^ AquaMax^®^ Fry Starter 200 and 300 three times a day. The protein levels are the same as above. The nutritional standards were based on the best available research and exceeded industry standards and recommendations [34,35].

### 2.11. Sample Collection and Genotype Analysis for F_1_ Generation

When F_1_ fish spawned in 2016 were 1.5 years old, pelvic fin samples were collected from 30 fish in each family. When F_1_ fish spawned in 2017 were 1 year old, pelvic fin samples were collected from 30 to 100 fish in each family. Surveyor mutation detection assay was conducted. The procedures were the same as described above.

To confirm the genotypes, Roche high-fidelity PCR amplification targeting cfGnRH gene was performed with 10 representative samples of DNA from each F_1_ family. Then, QIAquick PCR Purification Kit (QIAGEN, Germantown, MD, USA) was used to purify the PCR products. Next, the purified PCR products were sent to the Genewiz company (South Plainfield, NJ, USA) for sequencing.

### 2.12. Pleiotropic Effects

For P_1_ fish, the survival rate for embryos, 6-month-old fingerlings, 3-year-olds, and 4-year-old adult fish was measured. For F_1_ fish, survival rate and body weight for F_1_ 1-year-old and 1.5-year-old fish were measured. Since the 2016 F_1_ fish were generated without hormone treatment while the 2017 F_1_ fish were generated after hormone therapy, the different treatments and the environmental effect during the year need to be considered when evaluating the fish survival. Dead fish from a columnaris outbreak in one tank were collected. The reproductive behavior was observed through the bottom of overhead aquariums.

### 2.13. Statistical Analysis

Statistical analysis of mutation rate, survival rate, hatch rate and body weight were performed with R studio software (RStudio Inc., Boston, MA, USA). Mutation rate, survival rate, and hatch rate were compared utilizing Fisher’s exact test in cases of small sample size. Student’s *t*-test was performed to compare hatch rate and spawning rate between treatments and controls for P_1_ fish. Student’s *t*-test was also applied to compare survival rate and mutation rate between F_1_ families before and after hormone therapy and compare body weight in each F_1_ families. Shapiro-Wilk’s test was used to test the normality of the data. Comparisons of statistical significance were set at *p* < 0.05, and all data were presented as the mean ± standard error (SEM).

## 3. Results

### 3.1. Analysis of Mutation Efficiency and Plasmid Integration in the P_1_ Generation

The cfGnRH gene was successfully mutated in 27 of 51 6-month-old P_1_ fingerlings (52.9% mutation rate) based on Surveyor assay results, which showed three or five bands on the gel for both pelvic fin and barbel samples. The wild-type fish showed only one 550 bp band. The 350 bp band indicated that the cfGnRH gene was mutated at the expected site (Figure 1). Sequencing of cloned PCR products confirmed that the channel catfish were successfully mutated for the cfGnRH gene and included insertions, deletions, and substitutions, all of which were within the target TALEN cutting site. These mutations were in the GAP region of the cfGnRH gene (Figure 2A,B).

For the thirteen brood fish in 2016, the sequencing results showed one base of insertion at the targeted site was identified for one of four (25.0%) non-spawned males in one of twenty (5.0%) monoclonal colonies. Moreover, one base missense substitutions were identified for one of four (25.0%) non-spawned males and two of four (50.0%) spawned males in one of twenty (5.0%) monoclonal colonies separately for both sets of fish. No mutations were found for other P_1_ fish in limited picked monoclonal colonies, but the positive Surveyor Mutation Assay results suggested the potential mutation at the targeted site.

Neither CMV promoter region nor TAL repeats region were detected via PCR for any TALEN-cfGnRH mutant fish. Thus, none of the fingerlings carried the exogenous DNA (Figure 3A,B).

### 3.2. Spawning Experiments and Hormone Therapy

In 2016, three of five (60.0%) control pairs spawned with a 74.3% average hatch rate without LHRHa implantation. P_1_ mutants had spawning rates of 20% for females and 50% for males when mated with wild-type males and females, respectively; when there was no hormone therapy, the mean egg hatch rate was 2.0%, except for one cfGnRH mutated female that had a 66.0% hatch rate (Table 2 and Figure 4). The mean hatch rate of the mutant group was lower than the control group (*p* = 0.009). Three pairs of mutant channel catfish did not spawn after LHRHa hormone therapy with a dose of 90 μg/kg; however, it was late in the spawning season, and reabsorption of ova may have already initiated.

After establishing low fertility for GnRH gene-edited channel catfish in 2016, LHRHa hormone therapy of 100 μg/kg was implemented in 2017. Five of seven (71.4%) control pairs spawned with a 32.3% average hatch rate without LHRHa injection, while two of three (66.7%) control pairs spawned with a 56.5% average hatch rate with LHRHa injection in 2017. Spawning rate for female and male mutants mated together and receiving hormone therapy were 50.0% and 60.0% (one male spawned with two different females), respectively. The spawning rates of mutants were not different (*p* = 0.923) from controls without LHRHa injection, 71.4%, and controls with LHRHa injection, 66.7%. F_1_ embryo hatch rate for female and male mutants that received hormone therapy were both 72.1%, which was not different (*p* = 0.138, *p* = 0.615, respectively) from the non-injected control and the injected control, 32.3% without LHRHa injection and 56.5% with LHRHa injection (Table 2).

### 3.3. Analysis of Mutation Efficiency for the F_1_ Generation

Four F_1_ families were generated from mutant (M) fish and wild-type (Wt) fish in 2016, including 2016 GnRH-1 (Wt♀ × M♂), 2016 GnRH-2 (Wt♀ × M♂), 2016 GnRH-3 (Wt♀ × M♂) and 2016 GnRH-4 (M♀ × Wt♂). Three M♀ × M♂ F_1_ families were produced in 2017, including 2017 GnRH-1, 2017 GnRH-2, and 2017 GnRH-3. The Surveyor mutation detection assay results for mutated fish showed three or five bands on the gel (Figure 5). Each lane represents individual fish. The different patterns indicated mutations located at different sites. Sample No. 1–3 showed three bands, indicating that DNA double strands were cut at one site, so there is one mutation identified. No. 4–5 showed five bands, indicating that there is more than one mutation identified. The 1.5-year-old 2016 F_1_ families had 60.0%, 63.3%, 76.6%, and 66.7% mutation rates, respectively. The 1-year-old 2017 F_1_ offspring families had 53.0%, 46.7%, and 51.0% mutation rates, respectively.

The sequencing results show no mutation at the targeted site, and the targeted exon. However, base substitutions were found for F_1_ individuals at introns (Sequencing data not shown here). But these substitutions were not found for their corresponding P_1_ parental fish. For their corresponding P_1_ parent fish, single base substitution mutation was observed at the targeted site.

### 3.4. Pleiotropic Effects

For P_1_ fish in the TALEN treatment group, 200 eggs were double electroporated, and 76 of the 200 hatched with a 38.0% hatch rate. After 6 months, the fry survival rate of the TALEN cfGnRH group was 67.1%. For the control group, 84 of 200 eggs were double electroporated without TALEN plasmids with a 42.0% hatch rate. After 6 months, the fry survival rate was 75.0%. No significant differences were found between treatment group and control group for both hatch rate and survival rate (*p* = 0.475, *p* = 0.223). After three years, survival rates were 50.0% for mutant fish and 33.3% for non-mutant full-sibling fish. After four years, survival rates were 41.2% for mutant fish and 37.5% for non-mutant full-sibling fish in a recirculating system; while in the pond, the survival rates were 46.9% for mutant fish and 33.3% for non-mutant fish. No significant differences were found between mutants and non-mutants for 3-year-old P_1_ and 4-year-old P_1_ in the recirculating system and pond, respectively (*p* = 0.270, *p* = 1.000, *p* = 0.672) (Table 3). Although, the means were not different, the gene-edited fish had a consistently higher observed survival.

During courtship, sexually mature channel catfish males and females usually use their tail fins to cover each other’s eyes, and then their bodies quiver. The female deposits eggs into the water to form an egg mass, and the male discharges sperm almost immediately to fertilize the eggs. However, in this case, four of eight mutant males and four of five mutant females did not exhibit these normal reproductive behaviors, although all fish exhibited outstanding secondary sexual characteristics (Appendix A). In 2016, all the wild-type females were crossed with mutant males, but abnormal courtship for mutant males was observed through the bottom of the transparent aquaria. The mutant males were less willing to get close to the females, and they did not use their tail fins to cover female’s eyes or quiver their bodies. Some wild-type females laid unfertilized eggs without generating an egg mass. LHRHa hormone therapy resulted in good spawning and hatch rates for mutants in 2017. Three of five (60.0%) mutant males and three of six (50.0%) mutant females exhibited normal reproductive behavior. Their spawning rate and hatch rate were not significantly different from controls (*p* = 0.923).

For F_1_ fish, four families were obtained in 2016 before LHRHa hormone therapy, three families were obtained in 2017 after 100 μg/kg LHRHa hormone therapy. The mutation rates of the families obtained before hormone therapy with a mean of 66.7% were significantly higher than the families obtained after hormone therapy with a mean of 50.2% (*p* = 0.013) (Table 4). No significant difference in body weight between mutant fish and non-mutant fish were found in each family (*p* = 0.826, *p* = 0.598, *p* = 0.749, *p* = 0.296, *p* = 0.321, *p* = 0.563, *p* = 0.927). After evaluating the body weight in each family, an acute columnaris (*Flavobacterium columnare*) infection occurred in one family, 2017 GnRH-1. The mutation rate for this family was 53.0% before the infection. Fifty-six dead fish were collected in one tank, 36 of 56 (64.3%) dead fish were mutant fish, no significant difference in mutation rate between dead and alive fish was found in this family (*p* = 0.816). Body weight was not significantly different between mutant fish and non-mutant fish (*p* = 0.857) for these dead fish (Table 4).

## 4. Discussion

Mutations were detected at the targeted site of the cfGnRH gene. Different types of mutation were identified, including base deletion, substitution, and insertion with the total number of base changes ranging from 1 to 5. Theoretically, base deletion and insertion could shift the codon sequence, altering the open reading frame, changing the downstream amino acid sequence, and increasing the probability of premature stop codons. The base substitution could introduce a stop codon or change a codon to one that encodes a different amino acid and cause the change in the protein produced [13]. In the P_1_ generation, the mutation rate of TALEN-exposed 6-month-old fingerlings was 52.9%. The spawning rate and reproductive behavior of the P_1_ mutant fish and the hatch rate of their offspring were suppressed. The spawning rate of mutant fish was lower than wild types, especially for ovulation rate when they were mated without hormone-induced spawning. Average egg hatch rate was 2.0% except for one cfGnRH mutated female that had a 66.0% hatch rate. Hormone therapy restored the fertility of mutant P_1_ fish in 2017. No pleiotropic effects on body weight and survival rate were detected in mutant fish. This is the first sterilization achieved using TALEN technology targeting the GAP region in an aquaculture species.

TALENs have been applied as an effective method of mutagenesis in a variety of species through microinjection. TALEN application resulted in mutation rates ranging from 13% to 67% in mice [30]. Recent large-scale studies have reported TALEN mutation efficiencies between 3% and 95% [36,37]. The mutation rate via electroporation for the cfGnRH gene, 52.9%, was relatively high compared to other studies through electroporation. Two to three hundred one-cell-stage embryos can be microinjected in one batch of eggs [38], while at least four hundred one-cell-stage embryos can be electroporated in one batch of eggs. Electroporation was an effective and simple technique to utilize TALEN technology to induce mutations in channel catfish, and this can also be easily adapted for other fish species. There was no evidence for adverse off-target effects. These results appear promising for application in large-scale, commercial aquaculture.

Microinjection is usually used to transfer the mRNA into cells in previous studies. In channel catfish, designed single guide RNA of cfGnRH worked effectively at the targeted site with 50% mutagenesis. However, embryo hatch rates ranged from 0 to 42.4%, which is lower than electroporated fish in this study (72.1%). Microinjected embryos had low hatchability, when compared to their nCTRL siblings (75.8%). Injected siblings had low hatchability also (11.8%), which indicated that the low hatchability may be due to the microinjection procedures since all embryos were full-siblings, exposed to the same handling stress, and reared using the same environmental conditions [39].

In this study, TALEN plasmids were transferred instead of mRNA into the sperm and early embryos with double electroporation and successfully mutated the channel catfish GnRH gene. Electroporation can work as an alternative approach for delivering TALENs. The technique for animal knockout system by electroporation allows easy, rapid and high throughput generation of engineered animals. Electroporation and microinjection were both utilized to guarantee mutant transgenesis in channel catfish. Whether to choose electroporation or microinjection could be decided by the aim of the study [40], and for large-scale application, electroporation should be superior because of its simplicity and convenience. Introducing plasmids has advantages compared to introducing mRNA to induce mutations. Plasmids are easier to use and avoid degradation problems, and they require less time, effort and money.

Targeting only the hypothalamic form of GnRH can lead to sterilization. In medaka (*Oryzias latipes*), none of the female GnRH1 gene-knockout fish could ovulate, while all male GnRH1 gene-knockout fish were fertile [41]. In zebrafish, the GnRH3 gene-knockout fish are still fertile, displaying normal gametogenesis and reproductive performance in males and females [42]. The loss of both GnRH2 and GnRH3 isoforms resulted in no major impact on reproduction, indicating that a compensatory response, outside of the GnRH system was evoked [43]. Our results in channel catfish are different from both zebrafish and medaka. The spawning rate of mutants was lower than wild types, especially for ovulation rate when they were mated without hormone-induced spawning. Average egg hatch rate was 2.0% except for one cfGnRH mutated female that had a 66.0% hatch rate. In this study, channel catfish possess two forms of GnRH—cGnRH-ΙΙ and cfGnRH. The former is commonly accepted as a neuromodulator or plays a role in sexual behavior, while the latter is the hypothalamic GnRH form and considered to be responsible for the release of gonadotropin, playing a critical role in sexual maturation in channel catfish. So, only the cfGnRH gene was targeted in this study.

The P_1_ brood stocks were mostly mosaic. In some cases, dominant mutations may be associated with a loss of function. In some cases, two copies of a gene are required for normal function, so that removing a single copy leads to a mutant phenotype. In other cases, mutations in one allele may lead to a structural change in the protein that interferes with the function of the wild-type protein encoded by the other allele. A melanocortin-4 receptor gene homozygous mutation exhibited morbid early-onset obesity, and a heterozygote individual also had the possibility to have a milder overweight on mice [44]. As part of a multimeric complex, an abnormal variant might act as a “poison subunit”, rendering the whole complex nonfunctional, and such a mutant would behave genetically as an antimorph, disrupting proteasome activity in a dose-dependent manner [45,46].

In 2016, all the wild-type females were crossed with mutant males, but abnormal courtship was observed through the bottom of the transparent aquaria. These results indicated that two forms of GnRH may interact with each other to control the reproductive behaviors of channel catfish. Additionally, some wild-type females laid unfertilized eggs without generating an egg mass. The lack of normal reproductive behavior by mutant males could also have contributed to wild-type females responding inappropriately. LHRHa hormone therapy resulted in good spawning and hatch rates for mutants. Three of five mutant males and three of six mutant females exhibited normal reproductive behavior. P_1_ spawning rate was increased, and hatch rate was increased significantly compared with the results in 2016 before hormone therapy, excluding the three pairs in 2016 that did not spawn with hormone therapy likely due to the lateness of the spawning season and perhaps over-ripeness.

Cell toxicity can be an issue when the genome is edited. Cell death and apoptosis are most likely affiliated with off-target effects. Generally, TALENs are designed to target genes with high specificity and low cytotoxicity [47,48,49]. For the P_1_ generation, both the embryo hatch rate and fry survival rate of the TALEN treatment group was not different from the control group, which suggests that the TALEN plasmids and the nucleases had high specificity and only targeted the cfGnRH gene with little, if any, off-target mutations. After three to four years, no differences in survival were found between P_1_ mutant fish and their full-sibling controls cultured in recirculating systems and ponds further confirming this conclusion.

By 2025, aquaculture will need to increase by 350% worldwide to cover the impending seafood shortage, and by 2030 an additional 29 million tons of fish will be needed for human consumption [50]. Unfortunately, the United States is the leading global importer of fish and fishery products [51], and 91% of the seafood consumed by value is imported, resulting in a trade deficit, which is increasing every year and is now USD 11.2 billion annually. The United States is not in a position to take advantage of the opportunity of the expanding global market of aquaculture-grown fish, including catfish [52].

Exploiting fish genetics can greatly contribute to production and efficiency. The growth hormone (GH) transgenesis accelerated the growth of channel catfish [53] as well as in many other species, including mice (*Mus musculus*) [54], Coho salmon (*Oncorhynchus kisutch*) [55], Atlantic salmon (*Salmo salar*) [56], rainbow trout (*Salmo gairdneri*) [57], and common carp (*Cyprinus carpio*) [58]. β-actin-masou salmon desaturase transgenic channel catfish produced 25% more Ω-3 fatty acids compared to controls [59], and similar results have been found in common carp [32] and zebrafish [60]. Other traits such as disease resistance and resistance to abiotic factors have been improved via transgenesis in fish, including zebrafish [61] and channel catfish [52]. Gene editing has also altered traits such as growth in mice [62], yellow catfish (*Pelteobagrus fulvidraco*) [63] and channel catfish [64].

Because of public concern for environmental risk, transgenic and gene-edited fish are regulated by the Food and Drug Administration under the Federal Food, Drug, and Cosmetic Act and the Public Health Service Act [6,65]. For the commercial application of the genetic gains made through these two approaches, 100% reproductive confinement is necessary or should be the ultimate goal to prevent permanent environmental impact, negative or positive, of these genetically altered fish. Our initial results show that gene editing of the cfGnRH locus has great potential to achieve the goal of reproductive or biological confinement, the procedure is relatively straightforward and normal spawning protocols utilizing LHRHa application can temporarily restore fertility as needed. Next, cfGnRH channel catfish need to be generated that are also transgenic or edited for other genes to determine the feasibility of actual application of the technology. There is the possibility that gene editing of reproductive genes could decrease the benefit of growth hormone gene transfer or other genes such as was the case with triploid GH transgenic fish [66,67]. Until evaluated, potential epistatic effects of mutated alleles with transgenes or other mutated loci are unknown and need to be examined.

Ultimately, gene editing of reproductive genes coupled with hormone therapy could be used in a variety of fish, including any domestic genotype, invasive fish or transgenic fish. These technologies would minimize impacts on the natural environment to protect genetic biodiversity and ecosystems and increase the environmental friendliness of aquaculture and transgenic fish.

## 5. Conclusions

In the present study, TALEN plasmids were transferred into the sperm and early embryos with electroporation to induce mutations that would block the function of the cfGnRH gene in order to sterilize channel catfish, and then the fertility was restored in gene-edited sterile channel catfish through LHRHa hormone therapy. Mutants were sequenced and base deletion, substitution, and insertion were detected in P_1_ fish. No obvious effects on other economically important traits were observed after the knockout of the reproductive gene in the P_1_ fish. Growth rates, survival, and appearance between mutant and control individuals were not different. While complete knock-out of reproductive output was not achieved, as these were mosaic P_1_ brood stock, gene editing of channel catfish for the reproductive confinement of gene-engineered, domestic, and invasive fish to prevent gene flow into the natural environment appears promising.

## Figures and Tables

**Figure 1 biology-11-00649-f001:**
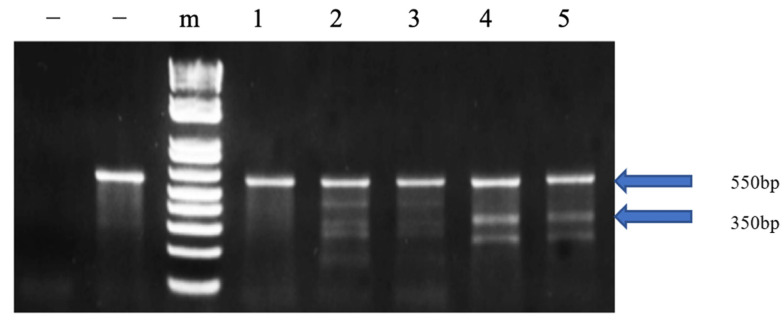
Identification of edited catfish-type gonadotropin-releasing hormone (cfGnRH) gene in P_1_ channel catfish (*Ictalurus punctatus*) using the Surveyor mutation detection assay. The left “−” indicates the negative control without template. The right “−” indicates the negative control with wild-type template; “m” indicates 1 kb DNA ladder; 2, 3, 4 and 5 are channel catfish with mutation; Lane 1 is a channel catfish without the mutation. Figure 1 was cropped from the full-length gels that were presented in Appendix A.

**Figure 2 biology-11-00649-f002:**
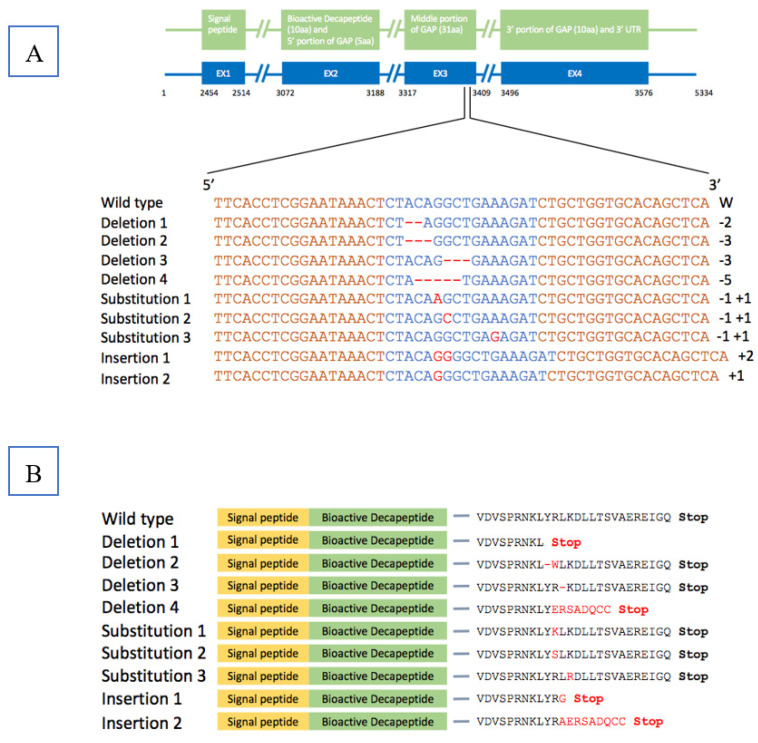
Nucleic acid sequences and corresponding predicted amino acid sequences of catfish-type gonadotropin-releasing hormone (cfGnRH) gene in wild-type channel catfish (*Ictalurus punctatus*) and after edited with transcription activator-like effector nucleases (TALENs). (**A**) EX1…EX4 indicate exon1…exon4. The wild-type channel catfish cfGnRH gene sequence is shown on the top. Sequences in orange are the target binding sites of the TALENs. Sequences (blue highlighted) in the middle portion of gonadotropin-releasing hormone associated peptide (GAP) are the expected cleavage sites of the nucleases. Red dashes and letters indicate the deletion, insertion/substitution of nucleotides. Numbers at the end of the sequences show the number of nucleotides deleted (−) or inserted (+) in the edited cfGnRH gene. (**B**) Predicted amino acid sequences with incomplete domain were due to frame-shift reading, resulting in a premature stop (red color) codon. Single amino acid substitutions or deletions (red color) were due to single nucleotide substitution mutation or three nucleotides deletion.

**Figure 3 biology-11-00649-f003:**
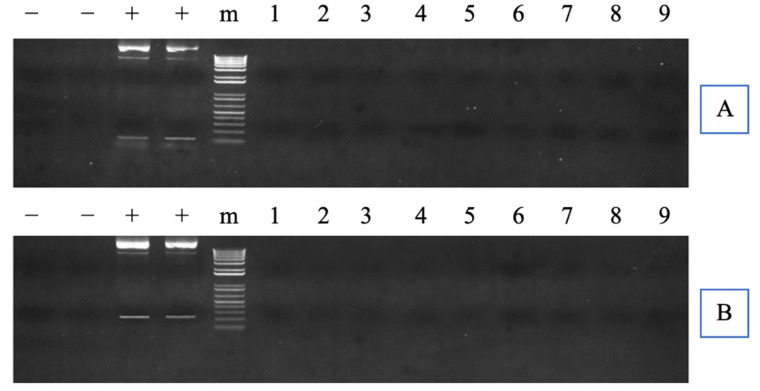
Polymerase chain reaction (PCR) inspection of potential transcription activator-like effector nuclease (TALEN) plasmid integration into channel catfish (*Ictalurus punctatus*) genome. In (**A**,**B**), the left lanes of “−” indicate the negative controls without template, while the right lanes of “−” indicate the negative controls with wild-type channel catfish DNA as a template; “+” indicates the positive controls with left and right TALEN plasmids as a template, respectively; “m” indicates 1 kb DNA ladders. Numbers represent samples from channel catfish individuals carrying mutated cfGnRH gene; the same number indicates the same individual in (**A**,**B**). (**A**,**B**) represent the PCR detection with different specific primers designed to amplify the cytomegalovirus (CMV) promoter region and the transcription activator-like (TAL) repeats region, respectively. (**A**,**B**) were cropped from the full-length gels that were presented in Appendix A.

**Figure 4 biology-11-00649-f004:**
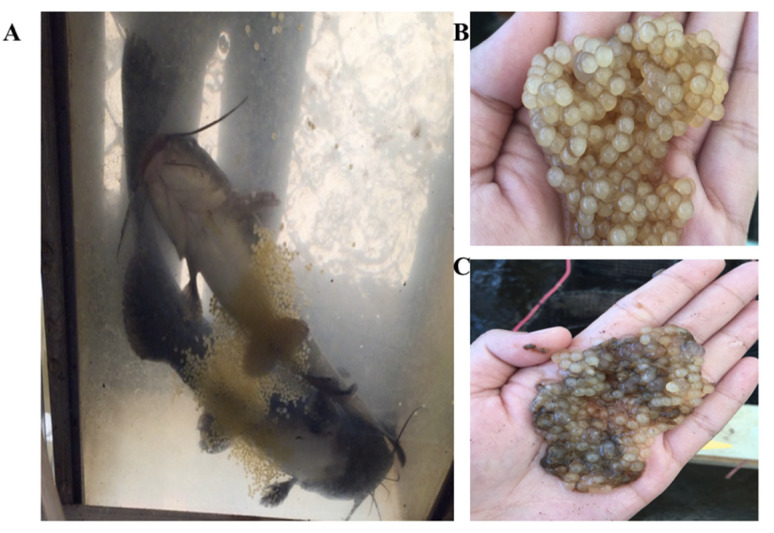
Images of the spawned P_1_ channel catfish (*Ictalurus punctatus*) in the aquarium, and the egg masses with high hatch rate and low hatch rate, respectively. (**A**) Showed the spawned P_1_ fish with normal behavior and egg mass. (**B**) Showed the egg mass with high hatch rate. (**C**) Showed the egg mass with low hatch rate.

**Figure 5 biology-11-00649-f005:**
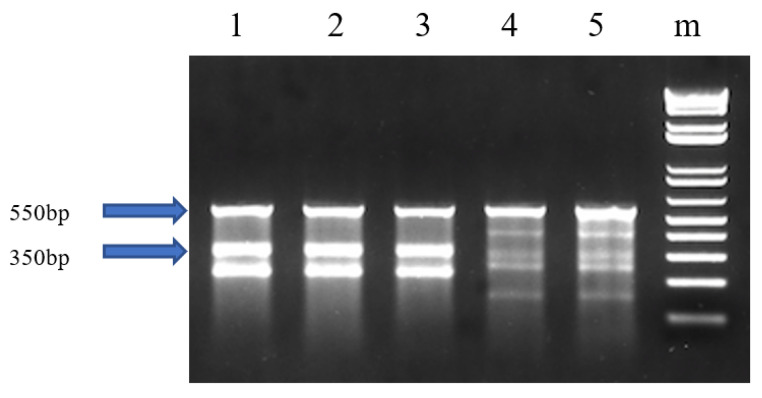
Identification of edited catfish-type gonadotropin-releasing hormone (cfGnRH) gene in F_1_ channel catfish (*Ictalurus punctatus*) using the Surveyor mutation detection assay. The negative control is not shown in this image. “m” indicates 1 kb DNA ladder; 1, 2, 3, 4 and 5 are channel catfish with mutation. Figure 5 was cropped from the full-length gels that were presented in Appendix A.

**Table 1 biology-11-00649-t001:** Primer sequences used to amplify the catfish-type gonadotropin-releasing hormone (cfGnRH), the cytomegalovirus (CMV) promoter and the transcription activator-like (TAL) repeat regions in channel catfish (*Ictalurus punctatus*).

Targeting Site	Sequence	Annealing Temperature	Product Length (bp)
cfGnRH Forward	5′-ATGGATGCTGTCTTTGTTTTCC-3′	60 ° C	550
cfGnRH Reverse	5′-CCACACGAAATAAAGGCAAAG-3′		
CMV promoter Forward	5′-AACAACAACGGCGGTAAG-3′	60 ° C	114
CMV promoter Reverse	5′-CCCATTATTGTTCGCGATTG-3′		
TAL Repeats Forward	5′-GCATGACGGAGGGAAAC-3′	60 ° C	215
TAL Repeats Reverse	5′-CCATTATTGTTCGCGATTGA-3′		

**Table 2 biology-11-00649-t002:** The spawning rate of P_1_ mutants and the mean embryo hatch rate of F_1_ channel catfish (*Ictalurus punctatus*) before and after 90 μg/kg luteinizing hormone-releasing hormone analog (LHRHa) hormone therapy in 2016, and after 100 μg/kg LHRHa hormone therapy in 2017. Two types of controls were used, including non-injected control (nCTRL) and injected control (iCTRL) with 90 μg/kg LHRHa solution. In 2016, 5 mutant females and 8 mutant males were paired with wild-type fish before hormone therapy. Only the wild-type females were implanted with 75 μg/kg LHRHa for induced spawning. In 2016, 3 mutant females and 3 mutant males were paired with each other with 90 μg/kg LHRHa hormone therapy. In 2017, 6 mutant females were paired with 5 mutant males with 100 μg/kg LHRHa hormone therapy (one male had two consecutive females). Hatch rate is the number of live embryos divided by the total amount of an egg mass in each family and multiplied by 100. Mean hatch rate data were presented as the mean ± standard error (SEM), which equals the number of eggs hatched divided by the total number of eggs for both mutant strains and wild-type strains. Spawned fish in the table means ovulated females or males with females that ovulated eggs.

		Before Hormone Therapy (2016)				After Hormone Therapy (2016)				After Hormone Therapy (2017)			
		Fish N	Spawned Fish N	Spawning Rate (%) ^a^	Mean Hatch Rate ± SEM (%)	Fish N	Spawned Fish N ^b^	Spawning Rate (%)	Mean Hatch Rate (%)	Fish N	Spawned Fish N	Spawning Rate ^c^ (%)	Mean Hatch Rate (%) ^d^
cfGnRH	F	5	1	20.0	66.0 *****	3	0	0	/	6	3	50.0	72.1 ± 0.07
	M	8	4	50.0	2.0 ± 0.01 *****	3	0	0	/	5	3	60.0	72.1 ± 0.07
nCTRL	F and M	5	3	60.0	74.3 ± 0.02 *****	/	/	/	/	7	5	71.4	32.3 ± 0.20
iCTRL	F and M	/	/	/	/	/	/	/	/	3	2	66.7	56.5 ± 0.34

* Hatch rate was significantly different between cfGnRH gene-edited group and nCTRL group before hormone therapy in 2016 (Student’s *t* test, *p* = 0.009). ^a^ Spawning rates were not significantly different among mutant females, mutant males, and control before hormone therapy in 2016 (Fisher’s exact test, *p* = 0.565). ^b^ Three pairs in 2016 did not spawn with hormone therapy likely due to the lateness of the spawning season and perhaps over ripeness. ^c^ Spawning rates were not significantly different among mutant females, mutant males, and nCTRL and iCTRL after hormone therapy in 2017 (Fisher’s exact test, *p* = 0.923). ^d^ Hatch rates were not significantly different between cfGnRH gene-edited group and nCTRL group, and between cfGnRH gene-edited group and iCTRL group after hormone therapy in 2017 (Student’s *t* test, *p* = 0.138, *p* = 0.615).

**Table 3 biology-11-00649-t003:** Comparisons of the embryo hatch rate and the 6-months-old fingerlings survival rate between P_1_ transcription activator-like effector nuclease (TALEN) plasmid treatment fish and the full-sibling controls. Comparisons of the survival rate of 3-year-old and 4-year-old P_1_ mutant channel catfish (*Ictalurus punctatus*) and their full-sibling treatment non-mutant fish cultured in recirculating systems and earthen ponds.

Treatment	P_1_ Embryo and 6-Month-Old Fingerlings				Genotype	3-Year-Old P_1_ Fish in 2016 (Recirculating)		4-Year-Old P_1_ Fish in 2017			
								Recirculating System		Pond	
	N Eggs	N Hatched	Hatch Rate (%) ^a^	Survival Rate (%) ^b^		N Fish	Survival Rate (%) ^c^	N Fish	Survival Rate (%) ^c^	N Fish	Survival Rate (%) ^c^
Electroporated with TALEN plasmids	200	76	38.0	67.1	Mutants	42	50.0	12	41.2	32	46.9
Electroporated without plasmids	200	84	42.0	75.0	Non-mutants	18	33.3	8	37.5	6	33.3

^a,b^ Hatch rate and survival rate were not significantly different between the treatment group and control group for P_1_ embryo and 6-month-old fingerlings (Fisher’s exact test, *p* = 0.475, *p* = 0.223). ^c^ Survival rates were not significantly different between mutants and non-mutants for 3-year-old P_1_ and 4-year-old P_1_ in the recirculating system and pond, respectively (Fisher’s exact test, *p* = 0.270, *p* = 1.000, *p* = 0.672).

**Table 4 biology-11-00649-t004:** Survival rate, mutation rate and mean body weight of the four families (2016 GnRH-1, GnRH-2, GnRH-3, and GnRH-4) of 1.5-year-old F_1_ offspring spawned in 2016 without hormone therapy, and the three families (2017 GnRH-1, GnRH-2, and GnRH-3) of 1-year-old F_1_ offspring spawned in 2017 with hormone therapy of channel catfish (*Ictalurus punctatus*). Mutant males and females were paired with wild-type fish to generate F_1_ offspring in 2016. Mutant males were paired with mutant females to generate F_1_ offspring in 2017. The fish in the 2017 GnRH-1 family died from columnaris (*Flavobacterium columnare*). The survival rate for this family was calculated after the outbreak of columnaris. Mean body weight data were presented as the mean ± standard error (SEM).

Different Families of F_1_ Offspring	Survival Rate			Mutation Rate *			Mean Body Weight (g) ± SEM of Mutant Fish ^a^	Mean Body Weight (g) ± SEM of Non-mutant Fish ^a^
	N Fish	N Fish Survived	Survival Rate (%)	N Fish Sampled	N Mutant Fish	Mutation Rate (%)		
2016 GnRH-1	30	14	46.7	30	18	60.0	164.5 ± 5.4	166.6 ± 7.8
2016 GnRH-2	30	12	40.0	30	19	63.3	154.3 ± 6.3	161.9 ± 12.4
2016 GnRH-3	30	22	73.3	30	23	76.7	70.7 ± 3.7	68.3 ± 6.3
2016 GnRH-4	30	16	53.3	30	20	66.7	154.9 ± 8.5	170.5 ± 11.8
2017 GnRH-1	300	224	74.7	100	53	53.0 ^c^	6.1 ± 0.2	5.9 ± 0.2
2017 GnRH-1 fish died from columnaris	/	/	/	56	36	64.3 ^c^	13.13 ± 0.5 ^b^	13.30 ± 0.8 ^b^
2017 GnRH-2	200	185	92.5	30	14	46.7	10.0 ± 0.7	9.4 ± 0.6
2017 GnRH-3	300	274	91.3	100	51	51.0	6.8 ± 0.2	6.7 ± 0.2

* Mutation rate was significantly different between F_1_ families without hormone therapy in 2016, and with hormone therapy in 2017 (Student’s *t* test, *p* = 0.013). ^a^ Body weight was not significantly different between mutant fish and non-mutant fish in each F_1_ families, including 2016 GnRH-1, 2016 GnRH-2, 2016 GnRH-3, 2016 GnRH-4, 2017 GnRH-1, 2017 GnRH-2 and 2017 GnRH-3 (Student’s *t* test, *p* = 0.826, *p* = 0.598, *p* = 0.749, *p* = 0.296, *p* = 0.321, *p* = 0.563, *p* = 0.927). ^b^ Body weight was not significantly different between the dead diseased mutant fish that died from disease and non-mutant fish in 2017 GnRH-1 F_1_ family (Student’s *t* test, *p* = 0.857). ^c^ Mutation rate was not significantly different between 2017 GnRH-1 family and the fish that died from columnaris in this family (Fisher’s exact test, *p* = 0.816). The mutation rate of the non-diseased fish was calculated before the outbreak of columnnaris.

## Data Availability

Not applicable.

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
