# Peer review of "Gene Editing of the Catfish Gonadotropin-Releasing Hormone Gene and Hormone Therapy to Control the Reproduction in Channel Catfish, Ictalurus punctatus"

_biology, 2022, doi:10.3390/biology11050649_

Round 1
Reviewer 1 Report
Manuscript ID: Biology-1657988
Gene Editing of the Catfish Gonadotropin-Releasing Hormone Gene and Hormone Therapy to Control the Reproduction in Channel Catfish, Ictalurus punctatus
In the current work, the author’s sought to eliminate the gonadotropin-releasing hormone through gene targeting to control fertility in catfish brood stock. I find the manuscript to be well written; I do however have some minor comments/suggestions that I hope will be addressed prior to publication.
- This is a minor point, I believe the figure numbering may be off, Figure 1 is referenced in the Figure 2 legend and Figure 1 isn’t mentioned in the main text at all and is in the back of the manuscript, please revise accordingly. Also in figure 5, figure 4 is referenced in the legend should say figure 5?
- Figure 3A, should there be red dashes in the nucleotide sequence of “Deletion 1” to represent the missing nucleotides.
- Table 1, while I don’t think is completely necessary, it might be interesting to describe the mutants used in each spawning experiment were all the male’s nucleotide deletions, substitutions or insertions; same for the females to give the reader a sense of which mutants were fertile and which ones were not without the hormone therapy. I ask this due to the lone mutated female with an almost normal hatch rate.
Reviewer 2 Report
In my opinion, this is a very interesting publication and it should be printed. Before that, however, it requires corrections. My detailed comments are contained in the text. to view them, open the file in Acrobat Reader. There is one thing I would like to point out: the word "mutant", which the authors use repeatedly. It is not wrong, but a human-induced mutation is described in the paper. Therefore, I suggest that either clarify this matter in the text or replace the word. In conclusion, I believe the authors will have no problem improving this work.

Round 2
Reviewer 2 Report
The MS was corrected due to my review. I recommend accept the MS at present form,